# Application of Nonlinear Models Combined with Conventional Laboratory Indicators for the Diagnosis and Differential Diagnosis of Ovarian Cancer

**DOI:** 10.3390/jcm12030844

**Published:** 2023-01-20

**Authors:** Tongshuo Zhang, Aibo Pang, Jungang Lyu, Hefei Ren, Jiangnan Song, Feng Zhu, Jinlong Liu, Yuntao Cui, Cunbao Ling, Yaping Tian

**Affiliations:** 1Department of Laboratory Medicine and Pathology, Jiangsu Provincial Corps Hospital of Chinese People’s Armed Police Force (PAP), Yangzhou 225003, China; 2Center for Birth Defects Prevention and Control Technology Research, Chinese PLA General Hospital, Beijing 100853, China; 3Third Department of Internal Medicine, Beijing Corps Hospital of PAP, Beijing 100027, China; 4Department of Laboratory Medicine, The Second Affiliated Hospital, Naval Medical University, Shanghai 200003, China; 5Department of Obstetrics and Gynecology, The First Medical Centre, Chinese PLA General Hospital, Beijing 100853, China; 6Department of Obstetrics and Gynecology, The 79th Group Army Hospital of PLA, Liaoyang 111000, China; 7Department of Laboratory Medicine, Characteristic Medical Center of PAP, Tianjin 300162, China

**Keywords:** laboratory diagnosis, machine learning, multiple indicator combinations, nonlinear model, ovarian cancer

## Abstract

Existing biomarkers for ovarian cancer lack sensitivity and specificity. We compared the diagnostic efficacy of nonlinear machine learning and linear statistical models for diagnosing ovarian cancer using a combination of conventional laboratory indicators. We divided 901 retrospective samples into an ovarian cancer group and a control group, comprising non-ovarian malignant gynecological tumor (NOMGT), benign gynecological disease (BGD), and healthy control subgroups. Cases were randomly assigned to training and internal validation sets. Two linear (logistic regression (LR) and Fisher’s linear discriminant (FLD)) and three nonlinear models (support vector machine (SVM), random forest (RF), and artificial neural network (ANN)) were constructed using 22 conventional laboratory indicators and three demographic characteristics. Model performance was compared. In an independent prospectively recruited validation set, the order of diagnostic efficiency was RF, SVM, ANN, FLD, LR, and carbohydrate antigen 125 (CA125)-only (AUC, accuracy: 0.989, 95.6%; 0.985, 94.4%; 0.974, 93.4%; 0.915, 82.1%; 0.859, 80.1%; and 0.732, 73.0%, respectively). RF maintained satisfactory classification performance for identifying different ovarian cancer stages and for discriminating it from NOMGT-, BGD-, or CA125-positive control. Nonlinear models outperformed linear models, indicating that nonlinear machine learning models can efficiently use conventional laboratory indicators for ovarian cancer diagnosis.

## 1. Introduction

Ovarian cancer has an insidious onset with no special symptoms or signs, and in more than 70% of patients, the cancer is not detected until late in the course of the disease and the opportunity for treatment is lost. Consequently, ovarian cancer has the highest mortality rate among gynecological tumors [1]. Effective screening tools for ovarian cancer remain lacking. Transvaginal sonography combined with serum carbohydrate antigen (CA) 125 is often used in clinical practice for the initial detection of ovarian cancer, but this approach has unsatisfactory sensitivity and specificity [2]. The results of transvaginal sonography often confuse benign pelvic masses with malignant ones [3] and are heavily influenced by the doctor’s experience. In contrast, peripheral blood testing has the advantages of being painless, minimally invasive, and quick, with higher acceptance and compliance; however, using CA125 is susceptible to false positives due to interference from benign tumors, inflammation, and hormone levels. Most previous studies [4,5] have shown that the area under the ROC curve (AUC) for subject operating characteristic curve is less than 0.8, which makes it difficult to meet clinical demand.

Diagnosis using a combination of multiple indicators is an effective method to address the shortcomings of single tumor markers. Among them, the generalized linear model, represented by logistic regression (LR) analysis and Fisher’s linear discriminant (FLD), tends to be more accurate than mechanical parallel or tandem diagnoses in ovarian cancer [6,7]. Laboratory tests combined with CA125 focus on serum tumor markers, such as human epithelial protein 4 (HE4), CA15-3, CA724, and carcinoembryonic antigen (CEA), while ignoring the potential value of more abundant blood routine, blood biochemistry, and other test data. As a branch of artificial intelligence (AI) technology, machine learning algorithms are rapidly emerging in the field of medical diagnosis given their ability to uncover hidden information and intrinsic associations among multiple variables. Medical AI systems grown out of machine learning have advantages over conventional statistical approaches in integrating enormous amounts of feature indicators, coping with missing data and iterating performance in multiple centers [8]. In recent years, with the development of liquid tumor biopsy technology, machine learning has mainly been used to analyze high-dimensional nonlinear data, such as mass spectra and gene sequencing of ovarian cancer samples [9,10,11]. Although data from routine tests was more easily available to all clinicians, taking into account a disease as heterogeneous as epithelial ovarian cancer with complex change pattern in laboratory indicators, prediction of ovarian cancer driven by conventional laboratory indicators has not been reported frequently. Moreover, past encouraging performance of AI systems for ovarian cancer concentrated mainly on classification of histologic subtypes [12,13] as well as prediction of survival, recurrence, and response to treatment [14,15,16]. The application of AI in screening ovarian cancer from a real-world cohort with diverse disease needs further investigation.

In this study, we made full use of the diagnostic information provided by various conventional tests in five classification models, both linear and nonlinear types, and compared their ability to diagnose ovarian cancer with a view of developing an economical and practical method for the intelligence-based diagnosis of ovarian cancer.

## 2. Materials and Methods

The design flow of the study is shown in Figure 1. This study was approved by the Ethics Committee of PLA General Hospital on 24 June 2021, under number S2021-364-01.

### 2.1. Study Subjects

Clinical data was retrospectively collected from patients who underwent physical examination at Chinese People’s Armed Police Force (PAP) from January 2010 to June 2019. The inclusion criteria were as follows: (1) discharge diagnosis confirmed by clinical signs, imaging, and pathology for patients with gynecological tumors and benign gynecological diseases; (2) if no histopathological examination was available, it must be consistently confirmed by two or more types of imaging evidence (computed tomography, magnetic resonance imaging, B-mode ultrasound, etc.); and (3) availability of laboratory test data at the time of first diagnosis and blood collection before treatment (including surgery, radiotherapy, and drug therapy). The exclusion criteria were as follows: (1) serious cardiac, pulmonary, hepatic, renal, hematologic, or other organ insufficiency; (2) combined acute or chronic infectious diseases; and (3) duration of pregnancy.

Finally, 901 study subjects (cohort 1) were enrolled and divided into the following groups: (1) The ovarian cancer group, comprising 224 patients with primary ovarian cancer, distributed in stages I–IV, based on the 2014 edition of the International Federation of Gynecology and Obstetrics (FIGO) ovarian cancer staging criteria, with the following pathological types: 105 cases of serous cystadenocarcinoma, 35 cases of mucinous cystadenocarcinoma, 36 cases of endometrioid adenocarcinoma, 19 cases of clear cell carcinoma, and 29 cases of non-epithelial ovarian cancer. (2) The remaining 677 women composed the overall control group. To test the ability of the model to identify related diseases in real-life application, the overall control group included the following three subgroups: 167 cases of non-ovarian malignant gynecological tumors, including 83 cases of cervical cancer, 69 cases of endometrial cancer, 8 cases of fallopian tube cancer, and 7 cases of choriocarcinoma; 382 cases of benign gynecological diseases, including 92 cases of ovarian cysts, 59 cases of endometriosis, 60 cases of pelvic inflammatory disease, 133 cases of uterine fibroids, and 38 cases of adenomyosis; and 128 cases of apparently healthy individuals in whom no benign or malignant gynecological disease was found on physical examination at the hospital physical examination center.

According to the same inclusion and exclusion criteria, 732 patients (cohort 2) were prospectively enrolled among women who attended the First Medical Center of Chinese PLA General Hospital and underwent physical examination between January 2021 and June 2022. These included the followings: (1) in the ovarian cancer group, 156 cases were distributed in FIGO stages I–IV, with the following pathological types: 103 cases of serous cystadenocarcinoma, 13 cases of mucinous cystadenocarcinoma, 16 cases of endometrioid adenocarcinoma, 11 cases of clear cell carcinoma, and 13 cases of non-epithelial ovarian cancer. (2) The control group comprised 576 individuals: 192 cases of non-ovarian malignant gynecological tumors, i.e., 81 cases of cervical cancer, 63 cases of endometrial cancer, 15 cases of fallopian tube cancer, and 33 cases of choriocarcinoma. Additionally, there were 175 cases in the benign gynecological disease group, including 41 cases of ovarian cysts, 33 cases of endometriosis, 14 cases of pelvic inflammatory disease, 52 cases of uterine fibroids, and 35 cases of adenomyosis. Moreover, there were 209 cases in the normal physical examination group.

### 2.2. Characteristic Indicators

To construct the models assessed in this study, we included 3 demographic indicators and 22 laboratory indicators.

#### 2.2.1. Demographic Indicators

From the many factors influencing ovarian cancer, the following three demographic characteristics were selected, which were confirmed by literature reports and were well-documented in the medical records: age, menstrual status, and number of pregnancies.

#### 2.2.2. Laboratory Indicators

From the tests commonly performed in the hospitals from which the cases originated, the following 22 hematological indicators belonging to 6 categories related to the occurrence or development of ovarian cancer were selected, based on published reports and experience accumulated in previous studies: (1) tumor markers, including CA125, CA15-3, CA72-4, CA19-9, CEA, alpha-fetoprotein, and squamous cell carcinoma-associated antigen; (2) blood routine markers, including absolute neutrophil value, absolute lymphocyte value, platelet count, and red blood cell distribution width; (3) coagulation function, including fibrinogen quantification and D-dimer quantification; (4) sex hormone markers, including chorionic gonadotropin (β-HCG); (5) biochemical routine markers, including albumin, globulin, C-reactive protein, and fasting blood glucose; and (6) lipid metabolism markers, including triglycerides, total cholesterol, low-density lipoprotein (LDL) cholesterol, and high-density lipoprotein (HDL) cholesterol. Some of the indicators were derived as more information-dense ratio forms for use in model construction, including neutrophil-to-lymphocyte ratio, platelet-to-lymphocyte ratio, albumin-to-globulin ratio, and LDL cholesterol-to-HDL cholesterol ratio.

If the same item was repeatedly tested, only the first test result prior to treatment was utilized. Missing test data were allowed but were limited to a less than 30% missing rate for each indicator in the ovarian cancer and control groups. The two hospitals involved in this study used HE4 as a second-line screening marker for ovarian cancer, not as a physical examination. Therefore, to avoid selection bias, HE4 was not included in the range of routine indicators used in this study.

### 2.3. Data Preprocessing

Missing-data interpolation was performed using the expectation-maximization method combined with multiple interpolations. For high abnormal values, truncating based on the marginal effect of diagnostic significance was performed (e.g., the upper limit of the range of CA125 was set to 1000 U/mL, and all values above 1000 U/mL were replaced with 1000 U/mL) [17]. Because the test data were obtained from two centers and the test reagents or instruments used were not uniform, cohorts 1 and 2 were each standardized: all indicators were transformed into dimensionless standard scores with a mean of 0 and standard deviation of 1 to ensure that the magnitude of data variation was comparable between different cohorts and tests. Using stratified random sampling, each group in cohort 1 was divided into a training set (2/3, 600 cases) and an internal validation set (1/3, 301 cases), according to a 2:1 ratio. All samples of cohort 2 were used as the external validation set.

### 2.4. Construction and Evaluation of the Diagnostic Models

#### 2.4.1. Unsupervised Clustering

Systematic clustering analysis and t-distribution random-domain embedding (t-SNE) dimensionality reduction were performed directly on samples using laboratory metrics from cohort 1. This was used to demonstrate the spatial distribution of samples after feature transformation for preliminary evaluation of the effectiveness of linear and nonlinear two-class treatment models in distinguishing the ovarian cancer group from the control group.

Systematic clustering is a linear iterative algorithm. In this study, the shortest-distance method was used for inter-class distance discrimination, and Euclidean distance was used to measure the similarity between samples. Starting from the closest sample, the samples were merged and then iteratively aggregated with other samples in order from near to far until all samples were merged into one large class, to obtain a spectral map of the clustering process.

t-SNE is a classical nonlinear dimensionality reduction algorithm. Different from cluster analysis, which divides the classes of samples by distance, t-SNE describes the distribution characteristics of samples by conditional probability and aggregates similar samples more uniformly. The t-SNE dimensionality reduction operation has a perplexity parameter of 50 and maximum of 1000 iterations, preserves three characteristic dimensions, and represents the distribution of samples using a three-dimensional scatterplot.

#### 2.4.2. Supervised Classification

With demographic characteristics and standardized processed laboratory indicators as input variables and dichotomous diagnostic results assigned to 0 (overall control group) and 1 (ovarian cancer group) as output variables, the correspondence between input and output was first established using the training set. The model was then used for prediction in the internal and external validation sets. The output results were expressed as predicted probabilities of continuous variables in the range of 0–1, to plot ROC curves and to calculate the summed net reclassification index (NRI) between the two models. The prediction probability of 0.5 was set as the negative–positive cutoff value, and the sensitivity, specificity, positive predictive value, negative predictive value, and accuracy were calculated for each diagnostic model. CA125 alone was used as a baseline for comparison to measure the diagnostic efficacy of multiple combination models. In accordance with the reference range of CA125 in the test report, CA125 of 35 U/mL was used as the negative–positive cutoff value.

(1)Generalized linear models

As general linear models, we used LR and FLD analyses.

In LR analysis, the backward stepwise regression method was used for input variable screening, and the variables retained in the last step and their regression coefficients were used to establish a binary LR equation.

In FLD analysis, the input variable screening method adopts a stepwise method, and each category of output variables automatically corresponds to a discriminant function, which can obtain the characteristic values F1 and F2, belonging to the ovarian cancer and overall control groups, respectively, and define F1/(F1 + F2) as the overall probability value.

(2)Nonlinear machine learning models

To develop nonlinear machine learning models, we used support vector machine (SVM), random forest (RF), and artificial neural network (ANN) algorithms.

For SVM, we selected a Gaussian radial basis as the kernel function, the coefficient of kernel function gamma of 2, and the coefficient of cost function of 0.2. We set RF to build a classifier model, containing 300 decision trees, each with 5 variables used for binary trees in nodes. For ANN, we selected a feedforward backpropagation neural network algorithm. The number of hidden layer neurons was 11 and learning rate was 0.1. The parameter optimization method algorithm was “rprop+”, and the activation function was “logistic”.

### 2.5. Statistical Analysis

The Statistical Package for the Social Sciences version 25.0 (IBM Inc., Armonk, NY, USA) was used for database management, descriptive statistics, and the construction of two linear diagnostic models (LR and FLD). R (version 4.2.1; https://www.r-project.org/, accessed on 5 June 2022) was used for diagnostic efficacy evaluation and for the construction of the three nonlinear diagnostic models (SVM (e1071 package), RF (randomForest package), and ANN (neuralnet package)). For the comparison of demographic indicators among the training, internal validation, and external validation sets, one-way analysis of variance was used for age and number of pregnancies, and the chi-squared test was used for menstrual status. The Delong test was used for AUC comparison between two models, and the Z-test was used to determine the significance of the NRI. *p* < 0.05 was considered a statistically significant difference.

## 3. Results

### 3.1. Clinical Baseline Characteristics of the Study Population

The demographic characteristics and stages of ovarian cancer at the two centers are shown in Table 1. The distribution of some demographic characteristics within the ovarian cancer, NOMGT, and BGD groups differed among the datasets (*p* < 0.05). Specifically, the training and internal validation sets of cohort 1 had similar characteristics, and the differences mainly originated from the external validation set. The different admission habits of the two hospitals and the imbalance between the groups were consistent with reality and facilitated in testing the stability of the diagnostic models in populations with different origins.

### 3.2. Visualization Comparison of Linear and Nonlinear Processing Modes

In the spectral map output of systematic clustering, different color blocks were mixed and arranged, indicating that the ovarian cancer group and each control group were scattered on different branches of the spectrum, and the differentiation effect was not satisfactory (Figure 2A), whereas in the three-dimensional space based on t-SNE downscaling, there was a clear trend of inter-group separation and intra-group aggregation between the ovarian group and each control subgroup sample. Although there was some local overlap of different color scatters, the global spatial segmentation boundaries were relatively clear in the spectral map (Figure 2B). In contrast, the nonlinear treatment pattern showed stronger potential for sample differentiation.

### 3.3. Internal Validation of Linear and Nonlinear Models

The classification results of the CA125-only model, two linear models, and three nonlinear models were compared. As shown in Table 2, by combining the evaluation metrics of AUC, sensitivity, specificity, positive predictive value, negative predictive value, and accuracy, the diagnostic efficacy was ranked, from highest to lowest, in the following order: RF, SVM, ANN, FLD, LR, and CA125-only. The traditional method of using CA125-only for diagnosing ovarian cancer performed poorly, with an accuracy rate of only 73.0%. The various linear and nonlinear models using multiple test results improved the accuracy to more than 80% and 90%, respectively, and the diagnostic efficacy of the nonlinear models was generally better than that of the linear models.

Among the linear models, the AUC of FLD was greater than that of LR (0.915 versus 0.859), with a significant difference (*p* < 0.001). Further calculation of the NRI revealed that FLD improved the correct classification ability by 18.9% over that of LR, with a significant difference (*p* < 0.001) (Figure 3B). Among the nonlinear models, the AUC of the RF was slightly greater than those of the SVM and ANN (0.989 versus 0.985 and 0.974, respectively), with no significant difference (*p* = 0.314 and *p* = 0.021, respectively), and the difference in NRI was also not statistically significant (*p* = 0.885 and *p* = 0.080, respectively), suggesting that the diagnostic efficacies of the three nonlinear models were similar (Figure 3).

### 3.4. External Testing of the Preferred Model among Linear and Nonlinear Models

The FLD and RF models, with the highest diagnostic efficacy for linear and nonlinear model types, respectively, were retained and were used on the external validation dataset for subsequent comparison. As shown in Figure 4, the AUC of the RF was greater than that of the FLD (0.898 versus 0.824), and the difference was statistically significant (*p* < 0.001). Compared with the results in the internal validation set, the AUCs of both RF and FLD decreased, which can be attributed to the differences in baseline characteristics between the two datasets. However, both models mentioned above maintained better diagnostic efficacy than that of CA125-only, consistent with their performance ranking in the internal validation set, showing that the models had stability and generalization ability.

### 3.5. Value of Preferred Nonlinear Models in Different Stages of Ovarian Cancer and Different Control Populations

The RF model with the best overall performance was used as a representative model for evaluating the performance of the established nonlinear models in data segmentation. The confidence interval (CI) of the AUC was obtained using the bootstrap method with 100 repetitions. Patients with ovarian cancer in the external validation set were classified into early (FIGO stages I–II) and advanced (FIGO stages III–IV) stages based on the FIGO staging criteria according to specific clinical needs. The AUCs of the RF model for the diagnosis of early and advanced ovarian cancer from the overall control group were 0.864 (95% CI, 0.807–0.992) and 0.915 (95% CI, 0.878–0.951), respectively, and the diagnostic efficacy for early-stage ovarian cancer was substantially improved compared with that of CA125-only (Figure 5).

In terms of clinical manifestations and index changes, the various categories of people constituting the control group showed varying degrees of similarity to patients with ovarian cancer, and the difficulties in differential diagnosis varied. In the external validation set, the AUCs of the RF model for independently discriminating ovarian cancer from non-ovarian malignant gynecological tumors, benign gynecological disease, and those with normal physical examination were 0.840 (95% CI, 0.795–0.885), 0.926 (95% CI, 0.896–0.956), and 0.939 (95% CI, 0.910–0.968), respectively, with a particular advantage in the ability to distinguish ovarian cancer from benign gynecological disease (Figure 6A–C). In addition, the AUC of the RF model reached 0.874 (95% CI, 0.841–0.916) when differentiating ovarian cancer from the control CA125-positive population (>35 U/mL), thus compensating for the lack of specificity of CA125, given the many factors affecting this marker, and thereby reducing misdiagnosis (Figure 6D).

## 4. Discussion

The present study constructed a total of five representative diagnostic models for ovarian cancer using readily available laboratory indicators and demographic characteristics as input variables. The findings highlighted three nonlinear machine learning models that had promising performance in comparison to two linear statistical models. The RF was the strongest performer out of the all the models, the AUC of which was 0.989 in the internal validation, and slightly lower at 0.898 in the external testing. A major strength of the nonlinear model was the general applicability in patients with different ovarian cancer stages as well as in various control groups, thus demonstrating its reliability in complex medical situations. It may supply a gap of cost-effective screening tools for ovarian cancer driven by routine laboratory data.

The new concept of “Clinlabomics”, which aims to maximize the use of routine data from medical laboratories, is gaining widespread attention [18]. The integration of highly standardized test data that is accumulated in hospital computer information systems and the expansion of their application to disease diagnosis or prognosis is an important way to reduce the cost of patient care and save medical resources. For example, serum globulin levels are generally elevated in the early stage of ovarian cancer and may be lower than the normal threshold in the late stage of immunosuppression. Moreover, CA125 levels in the normal population are influenced by multiple factors, such as age and menstrual status, and do not show a monotonic variation trend. However, the multivariate statistical methods commonly used in the medical field are based on linear logic. Although this approach certainly facilitates the mathematical description of complex problems, it is difficult to interpret and generalize the implied associations among dozens or even more characteristic indicators. Consequently, performance optimization might be achieved for diagnosis of ovarian cancer using a combination of multiple indicators. Machine learning is a complex and flexible nonlinear system that can automatically learn and identify relationships among variables. In this study, we performed nonlinear dimensionality reduction for cohort 1 using the t-SNE algorithm and demonstrated a clear spatial demarcation between the ovarian cancer and control groups, which suggested that nonlinear models would be more scientific and reasonable for mining diagnostic information from conventional biomarkers.

In the field of machine learning, more mature supervised learning algorithms include SVMs, RFs, and neural networks, and various algorithms have different characteristics and applications [19]. This study constructed the above three models simultaneously to evaluate the suitability of the data features and algorithms. The internal validation results showed that the AUCs of the three models were similar, and all exceeded 0.970, indicating satisfactory performance. The classification accuracy of the RF model was 94.7%, which was slightly better than those of SVMs and ANN; thus, the RF model was identified as the best model in our study. ANNs are currently the most widely used methods in the field of data mining. They achieve information processing by simulating the structure and function of the biological nervous system with good robustness and fault tolerance and can adequately approximate complex nonlinear relationships. The drawback of using neural networks lies in the need to set a large number of parameters, such as network topology and initial values of weights and thresholds. Our group [20] used a genetic algorithm to optimize the parameters of an ANN to improve the training speed and classification accuracy of a model. The AUC, sensitivity, and specificity of the method for diagnosing ovarian cancer were 0.948, 91.9%, and 86.9%, respectively, which were close to the diagnostic efficacy of the ANN used in this study, under the same sample grouping criteria. Qin et al. [21] constructed a multilayer perceptron with AUC, sensitivity, and specificity of 0.838, 72.60%, and 88.90%, respectively. These values were better than those obtained using single indicators, such as β-HCG, CA15-3, and CA125, but the model only incorporated six serum tumor markers and thus covered less information than in the present study. SVMs use kernel functions to map input variables to a high-dimensional space, finding the optimal hyperplane to segment the samples. These are good options for modeling nonlinear decision boundaries and mainly deal with high-dimensional binary classification problems. Paraskevaidi et al. [9] used infrared spectra of urine to identify ovarian cancer, and SVM using principal component analysis-filtered features performed well, with a sensitivity and specificity of 1 and 0.963, respectively, which were higher than those of two linear discriminant analysis algorithms. RF is a newly emerging integrated learning algorithm that consists of several randomly generated, independent decision trees that decide the type of outcome by voting. This approach is thus less susceptible to outlier interference, while avoiding the disadvantages of ANNs and SVMs that tend to fall into local optima and produce overfitting. Therefore, expectedly, RF performed better than the linear and other nonlinear models constructed in this study.

Different from most studies that only investigated patients with benign ovarian tumors or healthy populations as control groups [10,21,22,23], this study included several gynecological diseases with high incidence that are easily confused with ovarian cancer in the control group, to approximate real medical situations better and ensure the clinical applicability of the model. This increased the difficulty of model training and led to inconsistent results with previous similar studies. Kawakami et al. [23] constructed various machine learning models based on 32 clinical peripheral blood indicators for differentiating patients with epithelial ovarian cancer from those with benign ovarian tumors, and the comparison revealed that an LR model (AUC = 0.919) was inferior to RF (AUC = 0.968) and SVM (AUC = 0.939) models, but was slightly superior to an ANN model (AUC = 0.883). For the breakdown of pathological type to high-grade plasmacytoma, the diagnostic efficacy of the LR model was similar to that of the above machine learning models, and the advantages of the nonlinear combination model were not fully exploited. The Ovarian Malignancy Risk Algorithm index is a simple and efficient model for risk assessment of ovarian cancer development, consisting of a linear combination of two indicators, CA125 and HE4, and has a reported AUC of 0.96 by recent meta-analysis [24]. This diagnostic efficacy seems similar to that achieved in the present study after integrating more than 20 characteristic indicators. However, the result was obtained under the premise that the control group included only pelvic benign mass. Its serviceable range was narrower than the diagnostic models of this study with a diverse control group.

A diagnostic model must be independently validated to demonstrate its stability and generalizability. Therefore, an external validation set was created in this study by collecting cases from another hospital. The nonlinear model represented by RF still showed higher diagnostic efficacy than the linear model represented by FLD. This finding maybe because FLD does not restrict the distribution and variance of the original data and has a wider application than most linear models. However, it cannot overcome the influence of multicollinearity among the characteristic indicators. As the number of included test items increases, the phenomenon of multicollinearity becomes more prominent, which aggravates the loss of model accuracy. RF does not require the assumption of data prerequisites, and the model structure helps reveal non-parametric interactions among feature indicators. In particular, the use of RF models has been repeatedly reported in studies related to gut microorganisms, which vary greatly in composition and abundance [25,26]. RFs are expected to become mainstream models for dealing with biomarkers with ambiguous characteristic patterns.

To explore the application value of the diagnostic model in depth, this study tested the overall diagnostic efficacy of the RF model, but also performed separate analyses of its diagnostic ability in different stages of ovarian cancer. CA125, as the most commonly used ovarian cancer marker, has a significantly higher positive rate in advanced stages of ovarian cancer (84.10–92.40%) than in early stages (43.50–65.70%) [27]. Therefore, it is not optimal for early determination of the risk of ovarian cancer development [17]. The RF model was equally applicable to both early- and advanced-stage ovarian cancer and performed robustly. We also attempted to discriminate ovarian cancer from each independently stratified control group. Expectedly, RF was the best in distinguishing ovarian cancer from those with normal physical examination, but the scenario setting was remarkably simple and did not differ from that of CA125. When we used it to discriminate patients with ovarian cancer from those with benign gynecological conditions, the advantage of RF was more marked, as CA125 also increased to some extent under certain conditions, such as endometriosis and pelvic inflammatory disease, resulting in a decline in the discrimination ability of CA125 (AUC of 0.698 versus 0.926). When the difficulty level continued to increase, it seemed that the RF could not optimally differentiate between diseases that are also gynecological malignancies, which is consistent with what our group previously encountered when using ANNs for such differential diagnoses [20]. This is presumably because of the weak variation in the characteristics of peripheral blood indicators in various types of malignant gynecological tumors, the well-defined etiology of cervical cancer, and the symptoms of abnormal vaginal bleeding in endometrial cancer, which are well-established screening strategies that do not rely on peripheral blood tests [28]. Notably, the problem of false-positive CA125 tests severely limits the efficiency of screening for ovarian cancer [29], with a previous study noting that 9.6% of screened women had false-positive results and 6.2% underwent unnecessary surgery [30]. The present study was designed to provide a differential diagnosis for CA125-positive controls, and the RF model did not perform as well as it did in distinguishing patients with ovarian cancer from those with normal physical examinations or with benign gynecological disease. Nevertheless, its performance still substantially improved over that of CA125-only.

Limitations presented in data preprocessing should be considered. First, the diagnostic models mentioned above were based on a portion of assumed data by missing-data interpolation to increase the scale of the datasets and the number of input variables. As missing-data interpolation lacks a consensual standardized process, we could just learn from similar researches and explore a proper approach for imputing the missing data. Second, owing to distortion of clinical implication of outliers, a “physiologically based” normalization step was involved to benefit the accuracy of prediction. In this step, those indicators considered extremely high were truncated to within a range most relevant to the differentiation of patients, potentially leading to weaker interpretability and logical rigor in the modeling program. In conclusion, this study systematically compared the diagnostic efficacy of multiple linear and nonlinear models for distinguishing ovarian cancer in a domestic population, which has not been reported previously. Our study confirmed the value of using nonlinear combinations of conventional test results to provide clinical guidance in the diagnosis and differential diagnosis of ovarian cancer. This also provides a cost-effective approach for the screening and diagnosis of other complex diseases. In the future, the postoperative information of ovarian cancer cases should be tracked and used to evaluate the accuracy of nonlinear models in prognostic prediction of ovarian cancer, so that the application value of the nonlinear model can be enriched.

## Figures and Tables

**Figure 1 jcm-12-00844-f001:**
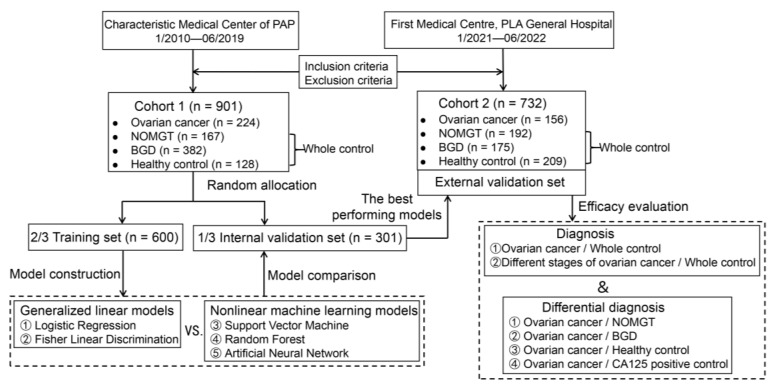
Flow chart of cases enrolled and model construction. NOMGT, non-ovarian malignant gynecological tumor; BGD, benign gynecological disease.

**Figure 2 jcm-12-00844-f002:**
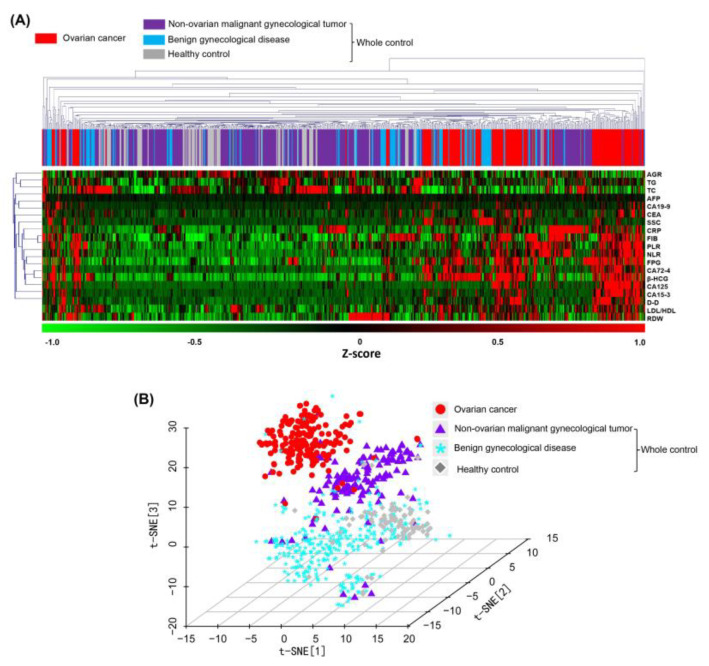
Demonstration of the differentiation effect of unsupervised clustering (four different colors represent ovarian cancer and three control subgroups). (**A**) Heat map of the differences in laboratory index levels with the spectral map of systematic clustering. AGR, albumin-to-globulin ratio; TG, triglycerides; TC, total cholesterol; AFP, alpha-fetoprotein; SCC, squamous cell carcinoma-associated antigen; CRP, C-reactive protein; FIB, fibrinogen; PLR, platelet-to-lymphocyte ratio; NLR, neutrophil-to-lymphocyte ratio; FBG, fasting blood glucose; D-D, D-dimer quantification; RDW, red blood cell distribution width. (**B**) Sample distribution in the space composed of three t-distribution random-domain embedding features.

**Figure 3 jcm-12-00844-f003:**
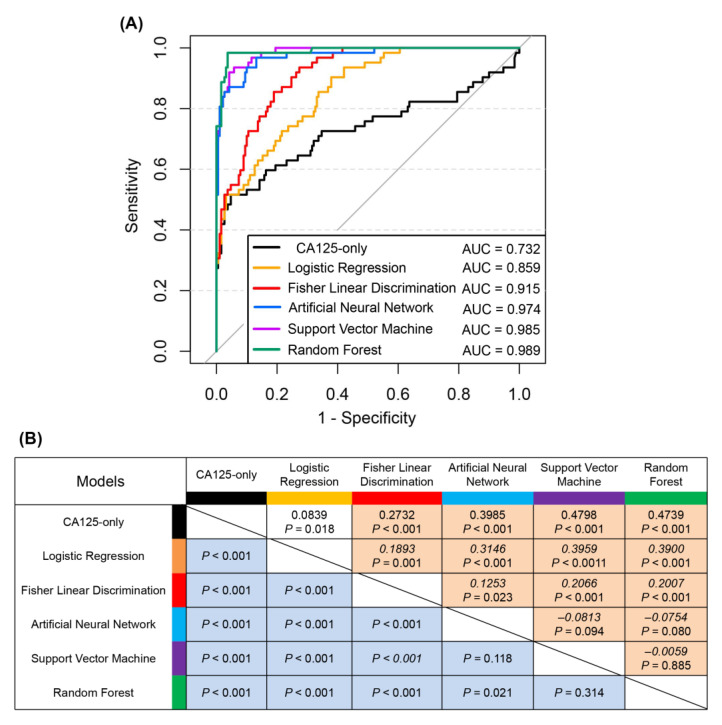
(**A**) Receiver operating characteristic curves for diagnosing ovarian cancer. (**B**) Multiple comparisons between each of the two models for diagnostic efficacy. Note: The datas in the orange cells are the net reclassification index (NRI) (%), and the *p*-value reflects the difference test between the two models. If the NRI is more than 0, it means that the model above the cell has better classification ability than the model on the left side of the cell, and vice versa if the NRI is less than 0. The datas in the blue cells are the *p*-values of the difference tests for the area under the curve of the two diagnostic models.

**Figure 4 jcm-12-00844-f004:**
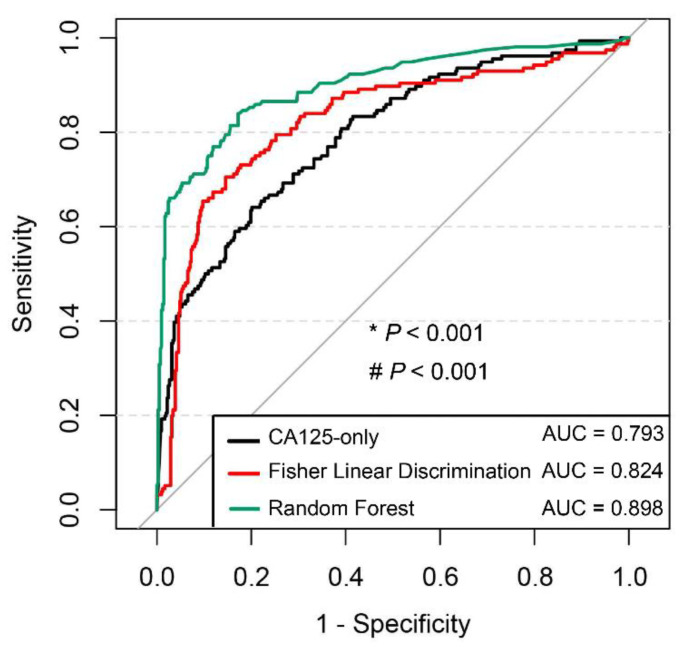
Receiver operating characteristic (ROC) curves of the preferred linear versus nonlinear models for the diagnosis of ovarian cancer. Note: * Area under the ROC curve (AUC) of random forest compared with AUC of Fisher’s linear discriminant model; # AUC of random forest compared with AUC of CA125-only.

**Figure 5 jcm-12-00844-f005:**
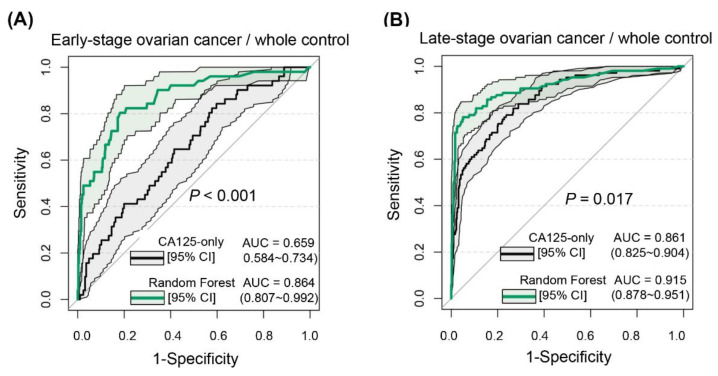
Receiver operating characteristic curves for different stages of ovarian cancer diagnosed by random forest. (**A**) Early-stage (International Federation of Gynecology and Obstetrics (FIGO) stages I–II) ovarian cancer versus overall control group. (**B**) Advanced-stage (FIGO stages III–IV) ovarian cancer versus overall control group.

**Figure 6 jcm-12-00844-f006:**
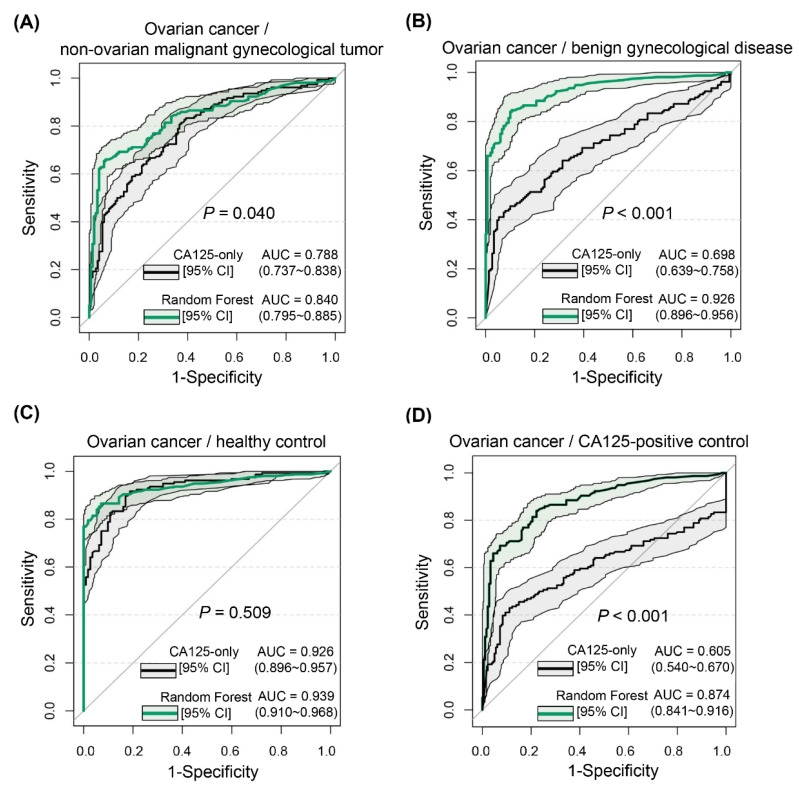
Receiver operating characteristic curves of random forest-based differential diagnosis of ovarian cancer versus different control populations. (**A**) Ovarian cancer versus non-ovarian malignant gynecological tumors. (**B**) Ovarian cancer versus benign gynecological diseases. (**C**) Ovarian cancer versus those with normal physical examination. (**D**) Ovarian cancer versus CA125-positive controls.

**Table 1 jcm-12-00844-t001:** Clinical information for each dataset.

Groups	Variable	Characteristic Medical Center of PAP		PLA General Hospital	*p*-Value
		Training	Internal Validation	External Validation	
Ovarian cancer	Count	149	75	156	—
	Age (years)	52.11 ± 14.27	52.69 ± 13.57	55.81 ± 12.52	0.043
	Gravidity	1.47 ± 1.15	1.44 ± 1.03	2.27 ± 1.52	<0.001
	Pre/post menopause	51/98	24/51	46/110	0.674
	FIGO stage				
	I/II	67	33	51	—
	III/IV	82	42	105	—
NOMGT	Count	111	56	192	—
	Age (years)	54.66 ± 11.87	52.66 ± 11.71	53.95 ± 12.05	0.595
	Gravidity	1.43 ± 1.05	1.64 ± 1.19	1.95 ± 1.47	<0.001
	Pre/post menopause	29/82	18/38	83/109	0.009
BGD	Count	255	127	175	—
	Age (years)	44.30 ± 12.62	46.33 ± 11.01	39.21 ± 9.59	<0.001
	Gravidity	1.82 ± 1.26	1.81 ± 1.37	1.50 ± 1.41	0.008
	Pre/post menopause	168/87	84/43	143/32	0.001
Healthy control	Count	85	43	209	—
	Age (years)	49.96 ± 13.92	47.26 ± 11.84	49.67 ± 9.78	0.117
	Gravidity	2.19 ± 1.62	1.97 ± 1.37	1.95 ± 0.96	0.165
	Pre/post menopause	50/35	29/14	103/106	0.055

FIGO, International Federation of Gynecology and Obstetrics; NOMGT, non-ovarian malignant gynecological tumor; BGD, benign gynecological disease.

**Table 2 jcm-12-00844-t002:** Diagnostic efficacy of various supervised classification models for ovarian cancer.

Model	AUC (95% CI)	Sensitivity	Specificity	PPV	NPV	Accuracy (%)
CA125-only	0.732 (0.644–0.819)	0.632	0.761	0.465	0.859	73.0
LR	0.859 (0.880–0.949)	0.615	0.862	0.593	0.868	80.1
FLD	0.915 (0.809–0.909)	0.849	0.811	0.607	0.940	82.1
ANN	0.974 (0.953–0.994)	0.821	0.970	0.892	0.938	93.4
SVM	0.985 (0.974–0.996)	0.917	0.948	0.857	0.969	94.4
RF	0.989 (0.978–1.000)	0.892	0.983	0.931	0.963	95.6

PPV, positive predictive value; NPV, negative predictive value; LR, logistic regression; FLD, Fisher’s linear discriminant; SVM, support vector machine; RF, random forest; ANN, artificial neural network.

## Data Availability

Not applicable.

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
