# Peer review of "Application of Nonlinear Models Combined with Conventional Laboratory Indicators for the Diagnosis and Differential Diagnosis of Ovarian Cancer"

_jcm, 2023, doi:10.3390/jcm12030844_

Round 1

Reviewer 1 Report

This article „Application of nonlinear models combined with conventional laboratory indicators for the diagnosis and differential diagnosis of ovarian cancer“ is the original articles from laboratory medicine aspect of diagnostics of ovarian cancer. The work has all the characteristics of originality with good statistics and presentation of results, as well as a discussion. references in a recent date and comparable. 

Author Response

We appreciate you very much for your positive comments on our manuscript. We will maintain these advantages you mentioned in the revised version.

Reviewer 2 Report

This article entitled, “Application of nonlinear models combined with conventional laboratory indicators for the diagnosis and differential diagnosis of ovarian cancer" demonstrated the diagnostic efficacy of various nonlinear machine learning and linear statistical models for diagnosing ovarian cancers. This study showed that RF could predict malignant ovarian tumors with an appreciably high accuracy compared with previous reports. The study is well written and the results are well presented and could be. However, there are some concerns as below.

1.      

It is rarely difficult to distinguish ovarian cancers from other gynecological malignancies in the patients after physical examinations. In this regard, the comparators in this study may be clinically inadequate.

2.

In the present study, three demographic indicators and 22 laboratory indicators were included. However, some hematological indicators, including chorionic gonadotropin, CA15-3, and CA72-4, were not evaluated usually in patients with gynecologic tumors. Therefore, this nonlinear model could not be acceptable in clinical application.

3.

The authors described “Malignancy Risk Algorithm index is a simple and efficient model for risk assessment of ovarian cancer development, consisting of a linear combination of two indicators, CA125 and HE4, and has a reported AUC of 0.91 (95% CI, 0.88–0.94) .” in line 412-414. However, recent meta-analysis revealed that the ROMA index were sensitivity: 0.90 (95% CI: 0.88–0.93), specificity: 0.91 (95% CI: 0.89–0.94), positive predictive: 0.90 (95% CI: 0.88–0.95), negative predictive: 0.93 (95% CI: 0.91–0.95), and area under ROC curve: 0.96. These results were similar to those of the present study. Therefore, the authors should discuss the superiority of this study compared to previous results.

4.

One of the novelty in the present study is to evaluate the diagnostic efficacy of linear and nonlinear statistic models for malignant ovarian tumors including non-epithelial ovarian cancers. It is interesting to evaluate the accuracy of these models in patients with non-epithelial ovarian cancers.

Author Response

Those comments are all valuable and very helpful for revising and improving our paper, as well as the important guiding significance to our researches. We have studied comments carefully and replied them point by point.

Reviewer 3 Report

Line 30 - glycoconjugate antigen 125 is only abreviated (CA125)
Line 62 - abreviations without introduction
Line 157 - Missing-value interpolation is a method that induce errors in data that no normalisation can cope. IN real life diagnosis is based on the available data and not on assumed data. The algorithms must use only existing data and if they need all variables a normal inclusion criteria must be their existence. What was the effect of tailoring CA125 on the sensitivity and specificity of CA125? This approach is to mathematical... we are not allowed to tailor patient findings.
Line 267 - the order of the algorithms must be consistent with table 2 that is introduced in the same paragraph, table 2 is not very easy to read in the present order and the same logical order must be used in figure 3.
Line 301 - FLD and CA125 curves are overlapping in several points and that is in contradiction with authors affirmation that "both models mentioned above maintained significantly better diagnostic efficacy than that of CA125-only", please correct.

Author Response

(The authors gave the same response as above.)

Reviewer 4 Report

This is an exciting research paper. 

However, a few suggestions are placed to further improve the manuscript.

Introduction:

Comment 1: Nicely written. However, one can elaborate more on the recent development in the field of AI/ ML for Ovarian CA and what changes it has made in the management of Ovarian CA. Also, the shortfall in this field should be highlighted which lead to the conceptualization of this research.

Method: 

Comment 2: Keeping the dual (retrospective and prospective) nature of the study, write the exclusion criteria separately and in detail. Why was the disease staging not taken into consideration while modelling/ taking characteristic indicators?

Results:

Comment 3: looks good

Discussion:

Comment 4: It is always better to start a discussion with the important findings of your result and the context in which it will be important or valuable than the already existing data.

Reference:

Comment 5: Some more references regarding usefulness/ previous works on Ovarian CA and AI/ML should be included.

Table and Figure:

Comment 6: looks good

Author Response

(The authors gave the same response as above.)

Round 2

Reviewer 2 Report

The revised manuscript is mostly improved in accordance with the reviewers' instructions. 

Author Response

We appreciate you very much for your positive comments on our correction in round 1. 

Reviewer 3 Report

Authors had made their corrections of the manuscript and I personally consider that a short version (one paragraph) of their response 3 must be included in the discussion as a drawback of the study.

Author Response

As reviewer suggested, we added one paragraph centred on drawback to the Discussion (page 16, lines 493–502).